# Monitoring Wind-Borne Particle Matter Entering Poultry Farms via the Air-Inlet: Highly Pathogenic Avian Influenza Virus and Other Pathogens Risk

**DOI:** 10.3390/pathogens11121534

**Published:** 2022-12-14

**Authors:** Armin R. W. Elbers, José L. Gonzales, Miriam G. J. Koene, Evelien A. Germeraad, Renate W. Hakze-van der Honing, Marleen van der Most, Henk Rodenboog, Francisca C. Velkers

**Affiliations:** 1Wageningen Bioveterinary Research, 8221 RA Lelystad, The Netherlands; 2De Heus Voeders B.V., 6717 VE Ede, The Netherlands; 3Department of Population Health Sciences, Faculty of Veterinary Medicine, Utrecht University, 3584 CL Utrecht, The Netherlands

**Keywords:** particle matter, plant, cobweb, faecal material, bird feathers, insects, wild aquatic avian species, *Campylobacter*, *Salmonella*, avian influenza virus, highly pathogenic avian influenza

## Abstract

Wind-supported transport of particle matter (PM) contaminated with excreta from highly pathogenic avian influenza virus (HPAIv)-infected wild birds may be a HPAIv-introduction pathway, which may explain infections in indoor-housed poultry. The primary objective of our study was therefore to measure the nature and quantity of PM entering poultry houses via air-inlets. The air-inlets of two recently HPAIv-infected poultry farms (a broiler farm and a layer farm) were equipped with mosquito-net collection bags. PM was harvested every 5 days for 25 days. Video-camera monitoring registered wild bird visits. PM was tested for avian influenza viruses (AIV), *Campylobacter* and *Salmonella* with PCR. Insects, predominantly mosquitoes, were tested for AIV, West Nile, Usutu and Schmallenberg virus. A considerable number of mosquitoes and small PM amounts entered the air-inlets, mostly cobweb and plant material, but no wild bird feathers. Substantial variation in PM entering between air-inlets existed. In stormy periods, significantly larger PM amounts may enter wind-directed air-inlets. PM samples were AIV and *Salmonella* negative and insect samples were negative for all viruses and bacteria, but several broiler and layer farm PM samples tested *Campylobacter* positive. Regular wild (water) bird visits were observed near to the poultry houses. Air-borne PM and insects—potentially contaminated with HPAIv or other pathogens—can enter poultry air-inlets. Implementation of measures limiting this potential introduction route are recommended.

## 1. Introduction

Wild avian species and poultry can be infected by highly pathogenic avian influenza A viruses (HPAIv), causing severe clinical disease and high mortality [1]. Since 2014, HPAIv of subtype H5Nx has been detected in commercial poultry, most frequently during the autumn and winter seasons (here referred to as bird-flu season) in the Netherlands [2,3,4]. The outbreaks in poultry were, except for the 2014–2015 epidemic, frequently preceded by findings of dead wild avian species infected with HPAIv near to the farm. These findings underline the importance of (in)direct contact of poultry with wild avian species. Therefore, the indoor housing of all free-range poultry was made mandatory during the bird-flu season. Despite the fact that indoor housing prevents direct contact between poultry and infected wild avian species, many poultry farms still became infected. At poultry farms, biosecurity measures are applied to prevent the introduction of infectious agents into the flock and to prevent farm-to-farm transmission. Commercial poultry farms should have biosecurity measures in place, formally expressed into a farm-specific written biosecurity plan, which can help adopt, improve and comply to high level biosecurity measures.

HPAIv-infected poultry can produce large quantities of virus that—stuck to particle matter (PM) such as feed dust and faecal particles—can become airborne [5] and can be transported by forced ventilation air from a house with infected poultry to the environment outside. This could lead to the potential windborne spread of the virus to other poultry farms [6,7]. The presence of airborne avian influenza virus (AIV) particles in the ambient air of wild bird wintering habitats has also been reported [8]. Similarly, these airborne particles could be generated by infected wild avian species near poultry farms [9]. A potential HPAIv-introduction pathway to poultry could be the wind-supported transport of PM via air-inlets. PM may consist of small wild bird feathers; scraps of plant and/or crop material; pieces of plastic, paper and cobweb and insects that are possibly contaminated by excretions or secretions from HPAIv-infected wild avian species. To our knowledge, no measurements are available on the nature of—and the quantity of—PM entering poultry houses via air-inlets. Therefore, the aim of our study was to measure and quantify the intake of PM through the air-inlet of two recently HPAIv-infected poultry farms. For these farms, based on epidemiological investigation, introduction of HPAIv via air-inlets was considered to be an option. PM and insects were tested for the presence of the influenza virus (IV) genome. In addition, we also tested for the presence of other pathogens as a proof of principle of the risk of the potential introduction of pathogens via the air-inlet of poultry houses. 

## 2. Materials and Methods

### 2.1. Poultry Farms

The study was executed on two separate farms with a very recent introduction of HPAIv subtype H5N1: a broiler farm and a laying hen farm for table eggs. On 7 and 14 November 2021, a suspicion of a HPAI-infection was raised for the layer and the broiler farm, respectively. Flocks were tested positive for HPAI H5 viruses and culled the subsequent day. The poultry houses of both farms were thoroughly cleaned and disinfected three times in a period of 45 days according to the standard procedures of the Netherlands Food and Consumer Product Authority (NVWA). The measurements started in these empty poultry houses 65 and 104 days after culling for the broiler and layer farm, respectively. Both farms were not yet populated with a new flock. In the layer farm, 25 sentinel chickens (placed to confirm freedom of infectious virus) were present in the poultry house during the measurements. Ventilation capacity in the empty poultry houses was set similarly to the settings for a normal-sized flock at the time of the flock cycle when the infection was diagnosed, to ensure an appropriate stream of air coming from the outside environment through the air-inlet into the poultry house.

#### 2.1.1. Broiler Farm

The broiler farm was situated in an area with 0, 1 and 9 neighbouring poultry farms in a 0-to-1 km, 1-to-3 km and 3-to-10 km radius, respectively, indicating the broiler farm was situated in a poultry-farm-low-density area. The broiler farm had four poultry houses; measurements were carried out in the only poultry house in which HPAIv-infection was detected. The house measured 75 m × 22 m × 6.5 m (length × width × height), with the length of the poultry house situated in a west-to-east direction. The prevailing wind was mostly from a western direction. The poultry house had a broiler density of approximately 34,000 broilers (regular fast-growing Ross 308, without outdoor range) at the start of the production round. The farm was situated close to the North Sea coast (approximately 10 km from the coast-line), in an area that is on the coast-line flyway of migrating wild aquatic avian species. The farm was closely surrounded by waterways (2–5 m wide) in which mallards (*Anas platyrhynchos*), Eurasian wigeons (*Mareca penelope*), Coots (*Fulica atra*) and Mute Swans (*Cygnus olor*) were observed after the farm was declared HPAIv-infected; a small (1 m wide, 20 m long) ditch filled with water was situated alongside the poultry house under investigation (distance to the poultry house: approximately 10 m). Pastures surrounded the farm, in which large quantities of Common Gulls (*Larus canus*) were observed during our study. 

#### 2.1.2. Layer Farm

The layer farm was situated in an area with 0, 0 and 17 neighbouring poultry farms in a 0-to-1 km, 1-to-3 km and 3-to-10 km radius, respectively, indicating the layer farm was situated in a poultry-farm-low-density area. The layer farm had three poultry houses, and measurements were carried out in the only house in which HPAIv-infection was detected in poultry. The house measured 43 m × 14 m × 6.5 m (length × width × height), with the length of the poultry house situated in a south-to-north direction. The prevailing wind was mostly from a western direction. The poultry house would occupy approximately 13,500 layers. The layer farm normally functions as a layer farm with an outdoor range. However, due to the national obligation to keep poultry inside due to the bird-flu threat, the layers for all houses were housed indoors at the time when HPAIv subtype H5N1 was introduced. The layer farm was situated close to the North Sea coast (approximately 10 km from the coast-line), in an area that is on the coast-line flyway of migrating wild (water) birds. The poultry farm was closely surrounded by small waterways (1 m wide); a large waterway (5 m wide) ran alongside the poultry free-range area in which Mallards, Eurasian wigeon, Coots, Tufted Ducks (*Aythya fuligula*) and Mute Swans were observed. The poultry farm was surrounded by pastures, in which several species of Geese (*Anser anser*, *A. albifrons*, *Branta leucopsis*) and a few Great Egrets (*Casmerodius albus*) were observed.

### 2.2. Air-Inlet Measurements

A total of 18 air-inlets-evenly distributed over both sides of the poultry house-inside the poultry house of both poultry farms were thoroughly cleaned and disinfected before they were covered (specifically for this study) by SKOV DA-1900 ventilation caps (SKOV A/S, Glyngøre, Denmark; www.skov.com, accessed on 12 November 2022) measuring 953 mm × 344 mm × 523 mm (length × width × height).

Each ventilation cap was equipped with a custom-made collection bag, consisting of polyester (mosquito) netting with a mesh-opening size of 1.4 × 1.6 mm (Figure 1). The ventilation caps and mesh collection bags were installed to be able to collect PM entering the poultry house with the normal air-stream bringing fresh air into the poultry house. A total of 21% (18 out of 86) and 37% (18 out of 49) of the air-inlets of the poultry houses under study were equipped with ventilation caps and collection bags on the broiler and layer farm, respectively. We choose to equip 18 air-inlets per poultry house with ventilation caps and collection bags primarily based on statistical considerations. There were no previous studies to rely on with respect to expected mean numbers and variation of PM items. To calculate the sample size for a mean obtained from a normal distribution, it is necessary to have an idea about the mean and its standard deviation (SD). Furthermore, an indication of the precision of the estimate is needed (L), along with the confidence in the estimate [10]. We used a conceivable SD = 4, L = 2 and 95% confidence, resulting in a calculated sample size of 16 air-inlets for measuring PM entering via air-inlets. The 5-times replication of the measurements over time (collection bags were harvested every 5 days for a total period of 25 consecutive days) would enhance the precision of the measurements. 

The collection bags were installed on 19 January on the broiler farm and 20 February 2022 on the layer farm. Each collection bag was first emptied upside-down by thoroughly shaking it for about 30 s over a large white cotton cloth on a table. Per air-inlet, all harvested items were identified by type, as artificial material (litter such as fabric fibre, glass, metal, plastic, etc.) or natural inanimate (cobweb, feather, wood, seed, leaf, pollen) or animate material (arthropod, arachnid, mollusc, etc.), counted and the size of the artificial and inanimate material was measured with a ruler (Figure 2). 

The collection bags were then swabbed on the inside thoroughly with a piece of polyester dust cloth (stofwisdoekjes; Kruidvat, Renswoude, Netherlands; www.kruidvat.nl, accessed on 11 November 2022), which improves the electrostatic particle adsorption. Per air-inlet, all harvested inanimate material and dust cloths were collected in a 50 mL Falcon laboratory tube (Scientific Laboratory Supplies, Nottingham, UK) and stored at −80 °C until further diagnostic analysis. The animate material was harvested from all air-inlets per measuring day together and stored in a 50 mL Falcon laboratory tube at −80 °C until further diagnostic analysis.

### 2.3. Video-Camera Monitoring

Video-camera monitoring (VCM) was installed to support the air-inlet measurements with temporal insight into all (agricultural) activities around the poultry house and the visits of wild avian species and other animals. A total of four Hikvision low light Turret 4 Mp video-cameras fitted with a fixed focus wide-angle 2.8 mm lens (view-angle of 98 degrees) (Hikvision, Hangzhou, China; www.hikvision.com, accessed on 21 October 2022) were installed at a height of about 2 m above ground-level on poles (Figure 3), covering the sides of the poultry houses with the air-inlets, the roof of the poultry house and the area around the poultry house. 

Video-cameras were connected to a TruVision NVR10 network video-recorder with HDMI/VGA video-output and a hard disk for the storage of video recordings (Interlogix, United Technologies Corporated, NC, USA; www.interlogix.com/truvision, accessed on 21 October 2022). The video-cameras were equipped with IR LEDs, enabling night-recording. Recording was performed at a speed of 2 frames/s, 24 h per day for the complete study period. 

Video recordings were replayed on a large “49” LCD monitor; all four video-camera images were displayed on the screen, with the possibility to focus and show only one camera image and even zoom in to obtain more visible detail. Recordings could be replayed at different speeds, and there was a possibility to archive snap-shots of specific video recordings. Specified characteristics of a wild bird visit were entered into an MS Excel database: date of visit; identification of visiting wild bird (Family, Order, Species); number of specific wild birds visiting; time (hh:mm:ss) of landing on the roof or area close to the poultry house; time (hh:mm:ss) of moving out of the area; estimated distance of location of wild bird to the poultry house. These data made a detailed characterization (length of time of a visit, the species involved) of each wild bird visit possible. A unique wild bird visit was defined as that in which one or more individuals of a wild bird species visited the area close to the poultry house (up to 10 m) or landed on the roof of the poultry house and stayed there for some time moving around; the length of time the bird remained in situ before departing was recorded. Wild birds visiting per observation day by species was defined as the total number of wild birds in visits per observation day by species [11]. The daily period of exposure of the poultry house or close area to the poultry house was calculated for each species by multiplying the number of counted individuals per unique wild bird visit by the total visiting time per species and sum them to a total for all wild bird visits per species per day. 

### 2.4. Testing for Pathogens

From each measuring day per poultry farm, a total of 10 samples (PM and the dust cloth; 50% of the total number of samples per measuring day; totalling 50 samples per poultry farm), were randomly selected for diagnostic testing. PM/dust cloth samples and all insect samples were tested for the influenza virus genome using matrix-gene real-time PCR, which detects all influenza virus subtypes, as described previously [2]. Furthermore, PM/dust cloth samples and all insect samples were tested for *Campylobacter* by PCR assay, based on the PCR as described by Josefsen et al. [11]. PM/dust cloth and insect samples were also tested for *Salmonella*, by an in-house-developed PCR test targeting the ttrRSBCA locus. In addition, all insect samples were tested for the influenza virus genome, Usutu virus and Schmallenberg virus, as described previously [12,13]. Insect samples were also tested for West Nile virus (WNV) with the VetMAX™ West Nile Virus Kit, WNPEXO50 (Thermo Scientific™, Waltham, MA, USA). Homogenization of the insects to prepare for diagnostic testing was performed as described earlier [14].

### 2.5. Descriptive Statistical Analysis

The distribution of the number and size of specific PM items entering the air-inlets was summarized by providing the median, 25th, 75th quantiles and the range using box-whisker plots. The median number of specific PM items entering the air-inlets on the wind-directed side of the poultry house of the layer farm (experiencing strong stormy conditions between two measuring intervals) was compared to that entering the air-inlets on the opposite side of the poultry house (wind-shadow side). Because the distributions of the number of PM items on the wind and wind-shadow side were not normally distributed, we used the nonparametric Kruskal–Wallis one-way analysis of variance test [15]. 

## 3. Results

### 3.1. PM via Broiler House Air-Inlets

The distribution of the number of PM per air-inlet per 5-day collection period in the broiler house is shown in Figure 4A. Cobweb and plant material were observed the most: a median of 1 cobweb or plant material per air-inlet per 5-day collection period. However, substantial variation was found between air-inlets, with up to 20 cobweb entries per 5-day collection period. No faecal matter or feathers of wild avian species were observed. Median PM size of cobweb, plastic and plant material ranged between 5 and 8 mm (Figure 4B). In exceptional cases, the PM of cobweb had a size up to 60 mm and the PM of plant material up to 40 mm.

### 3.2. PM via Layer House Air-Inlets

The distribution of the number of PM per air-inlet per 5-day collection period in the layer house is shown in Figure 4A. No feathers of wild avian species were observed. Cobweb and plant material entering via the air-inlet were observed the most. There is clear variation present: there were air-inlets with up to 60 cobweb and 20 plant material entries per 5-day collection period. The variation found was more substantial compared to the measurements on the broiler farm. This was the result of a huge storm (one day) between two measuring intervals (Beaufort wind scale 11: wind velocity of >100 km/h) that hit the western side of the layer house straight on. The median number of specific PM items in the netting of the air-inlets on that side of the layer house was significantly higher than that on the other side of the layer house, which was in the wind-shade: cobweb (*p* < 0.001); plant (*p* < 0.001); plastic (*p* = 0.004). In addition, during the storm, two pieces of dry faecal matter from outside were detected in a sample collected from one of the air-inlets. Based on macroscopic observation, the faecal material was not from a herbivore source (sheep, cattle, etc.), but probably from birds; the pasture area on the western side of the layer house would normally be used as a free-range area for the chickens (not during the bird flu epidemic because during that period there was obligatory indoor-housing of chickens), so it might be that the faecal material originated from chickens. The median PM size of cobweb, plastic and plant material ranged between 5 and 8 mm. In exceptional cases, the PM of plant material and cobweb had a size of up to 40 mm (Figure 4B).

### 3.3. Arthropods via Broiler House Air-Inlets

The distribution of the number of arthropods per air-inlet per 5-day collection period in the broiler house is shown in Figure 5. Of the arthropods observed coming through the air-inlet, the majority consisted of mosquitoes. A median of 12 mosquitoes was found per air-inlet per 5-day collection period, with in exceptional cases up to 65 mosquitoes per air-inlet per 5-day collection period. Furthermore, occasionally, other small insects or small spiders were observed.

### 3.4. Arthropods via Layer House Air-Inlets

The distribution of the number of arthropods per air-inlet per 5-day collection period in the layer house is shown in Figure 5. Also in this case, it concerned mainly mosquitoes: a median of 2 mosquitoes per air-inlet per 5-day collection period, but in exceptional cases up to 13 mosquitoes per air-inlet per 5-day collection period. Furthermore, occasionally, other small insects or small spiders were observed.

### 3.5. Wild and Domestic Animals Visiting the Direct Area around the Broiler House

The total number of wild birds visiting the area close to the broiler house is depicted in Figure 6A. The following wild bird species visited the area close to the poultry house or landed on the roof of the poultry house: Blackbird (*Turdus merula*), total exposure time (hh:mm:ss): 1:58:20; Black Crow (*Corvus corone*), total exposure time: 1:51:41; both species belonging to the order Passeriformes; Great Egret of the order Pelecaniformes, total exposure time: 0:30:01; and Mallards of the order Anseriformes, total exposure time: 5:13:11. Moreover, the area of the poultry house was visited on some occasions by mammalians such as house cats, house dogs and wild rabbits (*Oryctolagus cuniculus*).

### 3.6. Wild and Domestic Animals Visiting the Direct Area around the Layer House

The number of wild birds visiting the area close to the layer house is depicted in Figure 6B. The following wild bird species visited the area close to the poultry house or landed on the roof of the poultry house: Blackbird, total exposure time: 0:21:25; Black Crow, total exposure time: 0:14:29, both belonging to the order Passeriformes; Wood pigeon (*Columbus palumbus*) of the order Columbiformes, total exposure time: 0:31:12; Little black-backed gull (*Larus fuscus*), total exposure time: 9:58:07, and Oystercatcher (*Haematopus ostralegus*), total exposure time: 0:21:38, both species of the order Charadriiformes; and Mallards of the order Anseriformes, total exposure time: 4:53:16. Moreover, the area of the poultry house was visited on occasions by a house cat and wild rabbits.

### 3.7. Diagnostic Testing Results

PM/dust cloth and insect samples tested negative for the IV genome and *Salmonella*. Insect samples tested negative for Usutu virus, WNV, *Campylobacter* and Schmallenberg virus. However, there were indications of the presence of *Campylobacter* genome in the PM/dust cloth samples from both farms. A cut-off of a Ct-value of 36.0 was used to separate positive from negative samples in the PCR-test. On the layer farm, five out of 50 samples tested positive for *Campylobacter* DNA (mean Ct-value: 35.2, range: 34.7–35.8). However, in a further 31 of the samples from the laying farm (mean Ct-value: 37.6, range: 36.1–39.9) and eight of the 50 samples from the broiler farm (mean Ct-value: 38.4, range: 36.5–39.5) test results were inconclusive, with some Ct values very close to the cut-off value, suggesting the presence of low amounts of *Campylobacter* in these samples.

## 4. Discussion

During periods with enhanced risk of HPAIv introduction into poultry farms, e.g., in autumn–winter periods when HPAIv-infected wild avian species might be present in the environment, all poultry farms with free range in the Netherlands are ordered to house the poultry inside. Despite the fact that direct contact between poultry and infected wild avian species is prevented when poultry is housed indoors, many poultry farms become infected without conclusive evidence about the route of entry. At its best, a list of putative introduction pathways can be composed, mostly without a ranking. AIVs are shed from the cloacal, nasal, ocular and oral secretions from infected poultry and from wild avian species into the environment [1,16].

Airborne viral transmission-a mechanism by which a virus stuck to fine dust particles (faecal, feed) is transported through the air—between an infected poultry farm and a closely situated neighbouring farm with susceptible poultry is seen as a putative introduction pathway for HPAIv, based on analysis of data from earlier HPAIv-epidemics [6,7,17,18,19]. Under such circumstances and depending on the number of poultry being infected, a large titre of virus can be produced by an infectious poultry flock. However, in a field situation with HPAIv-infected wild avian species in the environment, the expected amount of virus and the concentration of virus produced by wild avian species-a combination of a low prevalence of infected wild avian species and a high degree of aerial dilution-is probably very low. Prevalence estimates of HPAIv-infected live aquatic avian species during the HPAI-epidemic season of 2014, 2014–2016 and 2016–2017 in the Netherlands were 5 per 10,000, 8.3 per 10,000 and 61 per 10,000 sampled live aquatic avian species, respectively [20,21,22]. The fact that pathogens may enter the poultry houses via air-inlets attached to particles carried by the wind was confirmed by our finding of the *Campylobacter* genome in some of the PM and dust cloth samples. For the layer farm in our study, we could not discern whether this could be linked to either the use of possibly *Campylobacter*-infected chickens when placed as sentinel chickens for detection of HPAIv introduction on the layer farm or airborne introduction of infection via contaminated PM and dust particles. However, in the broiler farm, which remained empty during the study, there were indications of *Campylobacter* presence in some of the PM and dust cloth samples. The PM and dust cloth samples collected in our study tested negative for the presence of *Salmonella* or IV genome. This does not exclude infection of a flock by HPAIv entering through the air-inlets. The absence of IV genome in the samples is not surprising as it is likely that (a) wind-supported entry of HPAIv via contaminated PM via air-inlets might be a very rare event, and (b) the probability of PM being contaminated and detectable with the available test methods by HPAIv might be low. 

In the biosecurity plan, areas and operational activities at the poultry farm that are at risk for introduction and spread of infectious agents are identified and appropriate measures designed to minimize that risk [23]. Examples of detailed and thorough biosecurity plans are available within the poultry industry [24], but the effectiveness of biosecurity measures heavily depends on the strict and consistent application of these measures by the farmer, personnel and visitors to the farm [25,26]. In a unique study, Racicot et al. [27] noted 44 different biosecurity violations using hidden cameras on eight poultry farms, especially concerning washing and disinfecting hands, and changing boots and coveralls between contaminated and clean areas. In our study, considerable numbers of avian influenza risk birds of the orders Anseriformes and Charadriiformes visited the area very close to the poultry houses, and therefore that area should be considered as potentially contaminated. It is therefore of paramount importance that the poultry farmer applies his biosecurity measures very strictly and consistently, especially with respect to a clear separation between the possibly contaminated area of the premises outside the poultry house and the clean area of the inside of the poultry house in which the poultry is housed.

The total number of wild birds visiting and wild bird exposure time of the area close to the poultry houses in this study during the winter months is in line with an earlier investigation on a free-range layer farm in the same northern region of the Netherlands [9]. HPAIv may be present in the dermis and epidermis of infected (wild) bird’s feathers [28]. Furthermore, HPAIv particles may accumulate onto the feathers of wild avian species, such as wild ducks [29,30]. The results of the study of Yamamoto et al. [29] indicate that the detached feathers of ducks infected with the HPAI H5N1 virus can also be a source of environmental contamination. In addition to the loss of feathers during moulting, flying or preening in live birds, the carcasses of dead wild (water) birds infected with HPAIv could be the source of loose feathers released when scavengers eat from the carcasses. Wild bird feathers, supported by wind, could possibly end up in a poultry house through air-inlets. Whether, how often, and in what quantity this could take place under field conditions is, however, unknown. It is also unknown whether these feathers still contain contagious HPAIv and can be a potential source of infection for the poultry, e.g., by ingestion. Despite substantial numbers of wild birds visiting the immediate surroundings of the poultry houses, no wild bird feathers were observed in the air-inlet collection bags and only in an exceptional case with stormy weather conditions two small pieces of faecal matter. It seems that direct wild bird material such as feathers and faecal matter are not entering the air-inlets by wind support. This does not rule out that this could happen, but it might be happening with an extremely low probability.

A potential HPAIv-introduction pathway from the outside environment as primary source to poultry inside a poultry building via air-inlets could be the wind-supported transport of PM from the farms’ surroundings, contaminated with HPAI-contaminated wild avian species excreta, such as faeces. In our study, we detected small amounts of PM that had entered the air-inlets which consisted of small scraps of plant and/or crop material, pieces of plastic, paper, wool and cobweb. During a storm, larger amounts of PM were found entering at the wind-directed side of the building. It should be noted that pathogens floating in aerosols, and wind-supported very fine dust and insects such as lice and mites of (sub)microscopic size, may have filtered undetected through the mesh of the collection bags.

Insects such as mosquitoes, blow flies and house flies are suggested as possible vectors of HPAIv. During a HPAI H5N1 epidemic on poultry farms in Japan, flies were collected from six sites within a radius of approximately 2 km from infected poultry farms [31]. HPAI H5N1 virus—identical to the virus strain derived from chickens, large-billed crows (*Corvus macrorhynchos*) and Black crows in the epidemic area—was detected in the intestinal organs, crop and gut of two blow fly species by RT-PCR, suggesting that blow flies can be a mechanical transmitter of HPAIv. Barbazan et al. [32] collected blood-engorged mosquitoes at HPAI H5N1-infected poultry farms during an epidemic in Thailand. Mosquitoes tested positive for HPAI H5N1 virus by RT-PCR. Salamatian et al. [33] conducted laboratory experiments with artificial feeding of house flies (*Musca domestica*) with a meal that contained low-pathogenic avian influenza (LPAI) H9N2 virus. External body washes and homogenates were tested with RT-PCR: both tested positive for H9N2. In a second experiment they showed that the persistence of LPAI H9N2 virus on external body surfaces and within the insects’ body were 24 h and 96 h, respectively. Overall, it was concluded that in these experiments house flies were able to obtain and preserve LPAI H9N2 virus, suggesting house flies might act as mechanical vectors.

We were surprised to observe considerable amounts of mosquitoes entering the air-inlets of our study farms. The above-mentioned investigations imply the potential of insects to act as (mechanical) vectors for between-farm transmission during an epidemic. In the Netherlands, since 2014, almost all HPAI-outbreaks are single primary introductions, with a source in the environment from possibly infected wild avian species around poultry farms. Blow flies and house flies may have contact with the carcasses of dead HPAIv-infected wild avian species or with faecal droppings from HPAI-infected wild avian species and may mechanically transmit the virus into the poultry farm. The probability that this will happen under natural circumstances is probably extremely low, but it cannot be ruled out. This emphasizes the idea that prevention of insects entering poultry houses, among others via air-inlets, might be important, not only for HPAIv but also for other pathogens. Stringent insect control in the poultry house needs attention and attachment of wind-break mesh nets in front of air-inlets is a possible solution to prevent the entrance of insects via the air-inlets of poultry houses. 

## 5. Conclusions

It is obvious that strict and consistent day-to-day application of biosecurity measures are needed to prevent the introduction of infectious agents into the flock and to prevent farm-to-farm transmission. However, our study also indicates that air-borne particle matter and arthropods—which could be potentially contaminated with HPAIv or other pathogens—may enter poultry air-inlets. Therefore, it is well-advised to limit this potential introduction pathway. New research is needed to understand how this can be performed in a proper way. The use of wind-break mesh netting might possibly make a contribution to limit this introduction pathway. 

## Figures and Tables

**Figure 1 pathogens-11-01534-f001:**
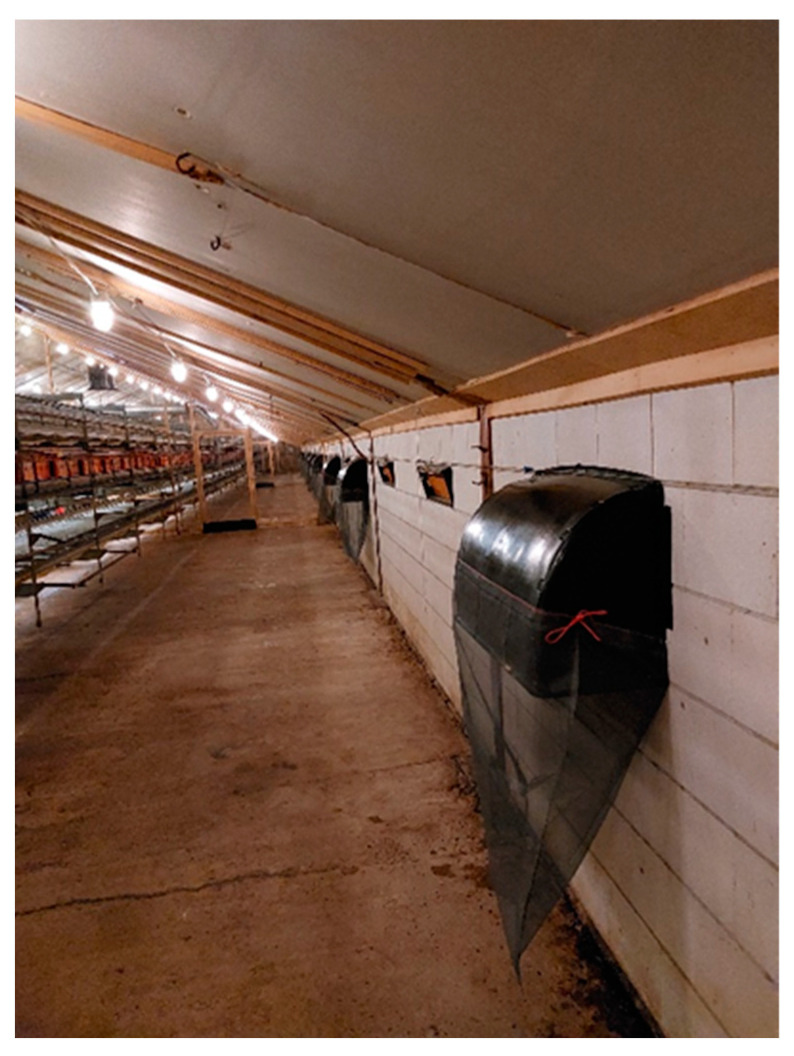
Air-inlets in the poultry house and air-inlets equipped with SKOV ventilation cap and collection netting.

**Figure 2 pathogens-11-01534-f002:**
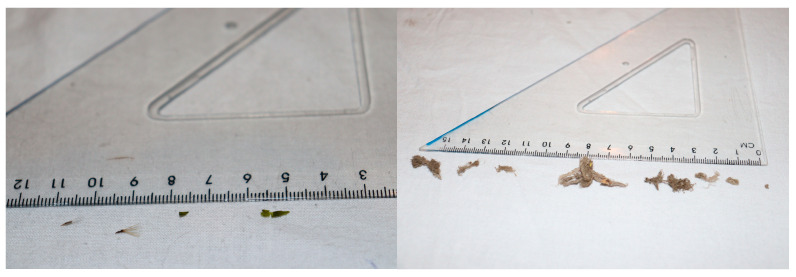
Examples of particle matter collected and measured with a ruler. In the left picture: seeds, plastic and leaf material. In the right picture: several pieces of cobweb with accumulated dust.

**Figure 3 pathogens-11-01534-f003:**
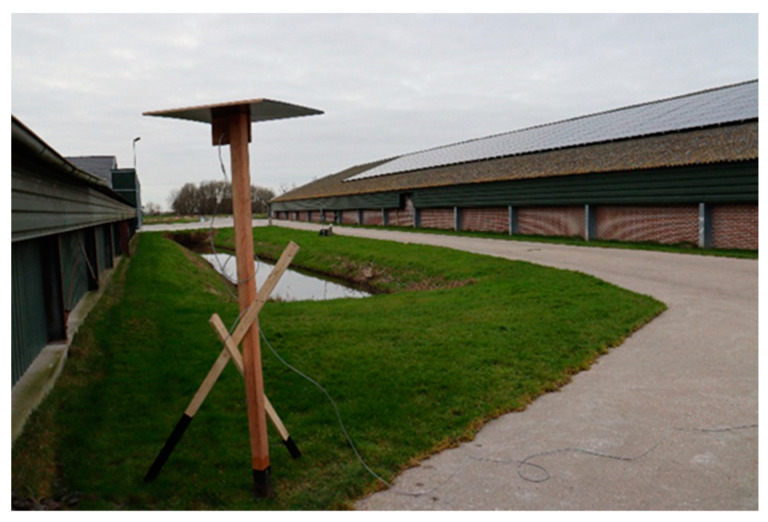
Video-camera installed to cover part of the area and roof of the poultry house under study.

**Figure 4 pathogens-11-01534-f004:**
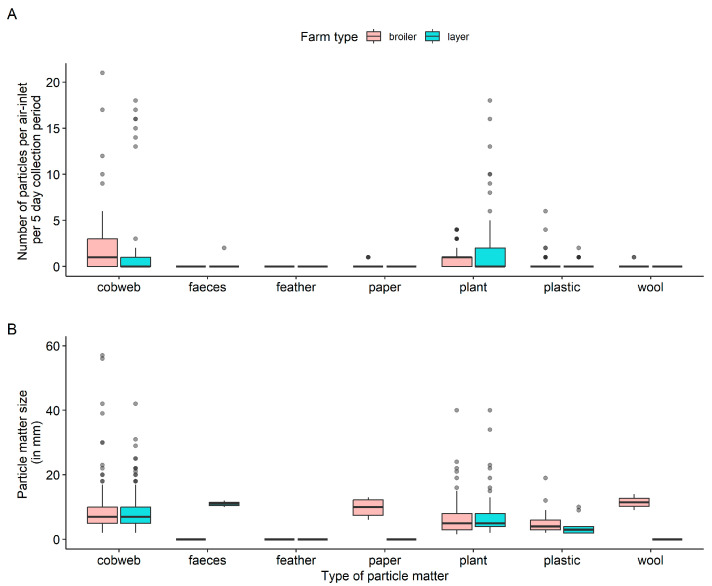
Distribution of the number of particle matter per air-inlet per 5-day collection period (total: 5 collection periods) in the broiler and layer house (**A**). Distribution of particle matter size from the air-inlets of the broiler and layer house (**B**). (fat dark line in the box: median; lower end of the box: 25% quantile; higher end of the box: 75% quantile; highest bullet or high end of the vertical line coming out of the box: highest value; lowest bullet or low end of the vertical line coming out of the box: lowest value). Note: The *Y*-axis of Figure 4A is cut off to prevent a disproportional layout with outlier cobweb measurement values up to 60 per air-inlet entries per 5-day collection period for the layer house.

**Figure 5 pathogens-11-01534-f005:**
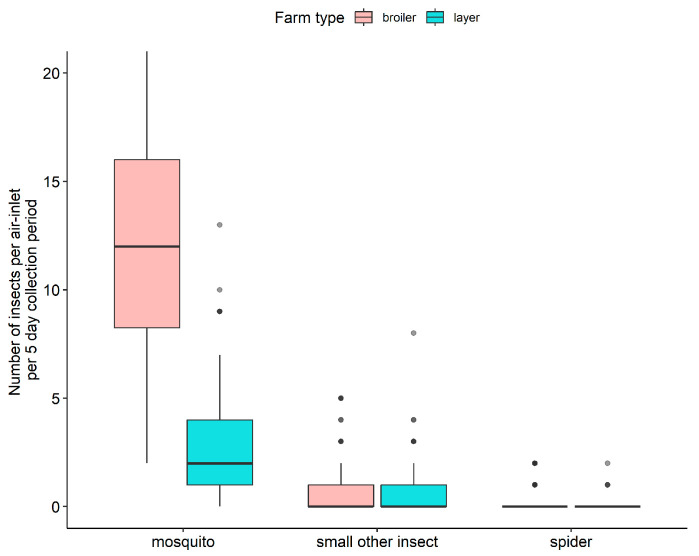
Distribution of the number of arthropods per air-inlet per 5-day collection period in the broiler and layer house; total 5 times 5-day collections per house (fat dark line in the box: median; lower end of the box: 25% quantile; higher end of the box: 75% quantile; highest bullet or high end of the vertical line coming out of the box: highest value; lowest bullet or low end of the vertical line coming out of the box: lowest value). Note: The *Y*-axis of Figure 5 is cut off to prevent a disproportional layout with outlier mosquito numbers up to 65 per air-inlet entries per 5-day collection period for the broiler house.

**Figure 6 pathogens-11-01534-f006:**
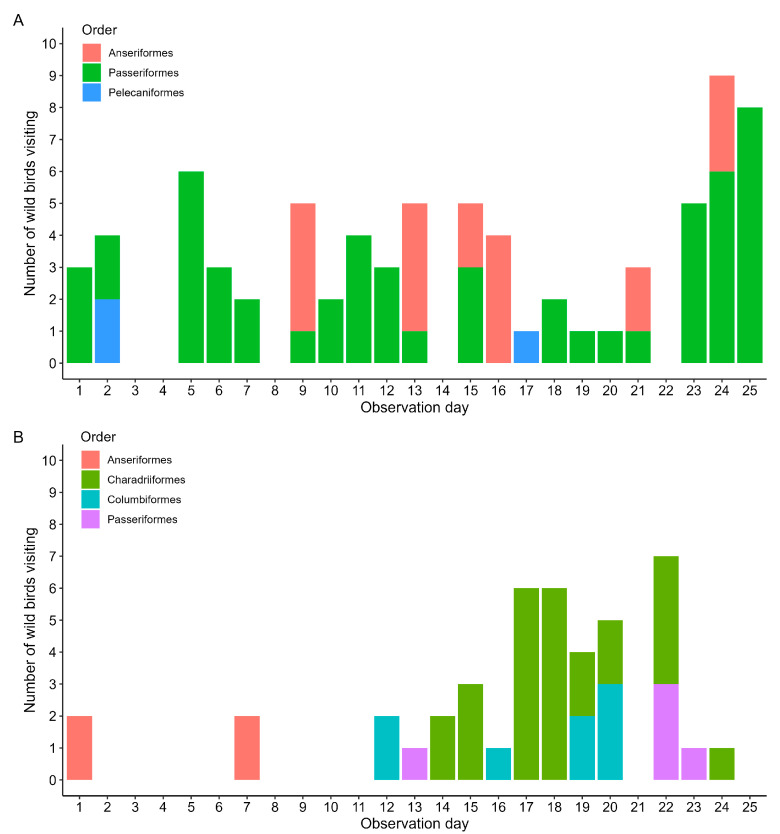
Distribution of the total number of wild birds visiting (by order and observation day) the area close to the broiler house (**A**) and layer house (**B**) during the study period.

## Data Availability

The data are available from the lead-author upon reasonable request.

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
