# Peer review of "Monitoring Wind-Borne Particle Matter Entering Poultry Farms via the Air-Inlet: Highly Pathogenic Avian Influenza Virus and Other Pathogens Risk"

_pathogens, 2022, doi:10.3390/pathogens11121534_

Round 1

Reviewer 1 Report

GENERAL COMMENTS

This manuscript investigates the potential role of wind-borne particle matter and insects in the transmission of highly pathogenic avian influenza viruses (HPAIV) and other pathogens in poultry farms. Airborne transmission of viruses attached to fine dust particles from an infected to a neighbouring susceptible poultry farms is commonly seen as a putative introduction. However, in the case of the Netherlands, where the study was conducted, most HPAIV outbreaks are primary introductions with an environmental source, possibly wild birds. In this case, airborne transmission of viruses attached to particle matters (wild birds feather or faeces) or via insects as mechanical vectors is suspected, especially in the context of HPAIV epidemics when indoor housing of poultry is made mandatory to avoid direct contacts between domestic and wild birds. Although airborne introduction to poultry is suspected, the nature and quantity of particle matter and insects entering poultry houses via air-inlets have never been measured before. The aim of this study was therefore to measure a quantify the intake of particle matters through the air-inlets of two chicken farms recently infected by HPAIV. Interestingly, the authors found substantial quantities of particle matters (mostly cobwebs and plants) and insects (mostly mosquitoes), which could represent a substantial risk of pathogen introduction. As expected, a high variability throughout the study period was observed, depending on the wind activity. Particle matters and insects were submitted to diagnostic tests for the presence of several viruses, including HPAIV, and also bacteria (Campylobacter and Salmonella). None of the samples tested positive for these pathogens, except several samples testing positive for Campylobacter. This support the hypothesis that airborne pathogen introduction via particle matters or insects in air-inlets of poultry farms is possible, although probably rare. However, a low probability of contaminated material, combined to a high frequency of entry of those material, may lead to a non-negligible risk of introduction. The authors therefore provide recommendations of potential measures to limit the introduction of particle matters and insects into poultry houses, which may be easy to implement and possibly efficacious.

The manuscript provides new results that will be of interest to the broad readership of Pathogens. It is well written and clear, and I have no major concern over its quality and the quality of the methods. However, I have identified a few points that need to be clarified before publication. I also made a few suggestions on the discussion section.

DETAILED COMMENTS

ABSTRACT

Line 16: I would add a second sentence to clarify the primary objective (and novel aspect) of your study, which was to measure the nature and quantify of PM entering poultry houses via air-inlets.

Lines 16-17: “Air-inlets of a recently HPAIv-infected broiler and layer farm were equipped with mosquito-net collection bags.” should read : “Air-inlets of two recently HPAIv-infected chicken farms (a broiler farm and a layer farm) were equipped with mosquito-net collection bags” for clarity.

Line 19: “and” should not be italicized

INTRODUCTION

Line 35: “has” should be replaced by “have” (several HPAI viruses)

MATERIAL AND METHODS

2.1 Poultry farms

It could be relevant, for each farm, to indicate the direction of prevailing wind (if known). This would help the reader understand how the poultry houses and their air-inlets are oriented compare to the wind.

Lines 112-116: it is not necessary to repeat the scientific names of the species you already provided before (lines 92-97); specify only the scientific names of species you mention for the first time.

2.2 Air-inlet measurements

Lines 119-122: does it mean that the air-inlets are usually not covered by this type of ventilation cap? If this is the case, it should be clarified, especially if there were no ventilation caps at the time when infection was diagnosed. Could these caps alter the air stream?

Lines 145-147: how did you choose the air-inlets that were equipped with ventilation caps and collection bags? Why did you choose to equip 18 of them in each farm?

2.3 Video-camera monitoring

Lines 210-212: “The daily period of exposure of the poultry house or close area to the poultry house was calculated for each species by multiplying the number of counted individuals by the total visiting time per species.” I am not sure this is correct, unless I misunderstood. Say one species made a visit with 10 birds for two minutes, and another visit on the same day with one bird for ten minutes. Do you calculate the daily period of exposure for this species by doing 10*2 + 1*10 = 30 minutes, or by multiplying the total number of counted individuals (11) by the total visiting time (12 minutes), i.e. 11*12 = 132 minutes? The former seems correct, but is not what I understood from your sentence. The latter seems wrong to me. Please clarify.

RESULTS

3.1. PM via broiler house air-inlet

Line 242: in Figure 3A, it seems that the upper value of the number of cobweb entries is more than 20.The point seems cut in half, so maybe you should increase the upper limit of the y-axis.

Lines 245-246: I suggest rephrasing: “In exceptional cases, PM of cobweb had a size of up to 60 mm and PM of plant materials up to 40 mm.”

For the layer house, you compared the median number of PM items between the wind-shadow side and the wind-directed side. I would be interested to see if there were any difference for the broiler house.

3.2. PM via layer house air-inlet

Lines 259-261: you mention up to 60 cobweb entries in the layer house, but this is not reflected in Figure 3A. Please clarify. The highest value of the number of plant material entries does not seem to be 20 in Figure 3A either.

3.3. Insects via broiler house air-inlet

Line 273: add a space between broiler and house

Lines 276-278: if the number of mosquitoes goes up to 65 per air-inlet per 5-day collection period, this should be somehow notified in Figure 4. To keep the figure readable, you could e.g. use a gap in the y-axis to show the upper limit of the vertical line of the first box-whisker plot, or e.g. at least notify in the figure legend that the highest value is 65 and was not represented in the figure.

3.4. Insects via layer house air-inlet

Line 289-291: the highest value for the number of mosquitoes is not 10 in Figure 4.

3.5. Wild birds visiting the direct area around the broiler house

Line 294-295: add a full stop after “Figure 5A”.

Lines 295-298: you don’t have to repeat the scientific names for wild bird species already mentioned earlier in the manuscript.

Besides the number of wild birds visiting the area close to the poultry farms, did you analyse the visiting times or the calculated daily period of exposure?

Figure 5: is this the number of unique visits (defined in the Methods as visits in which one or more individuals of a wild bird species visited the area) or the total number of wild birds visiting? Please clarify.

3.6. Wild birds visiting the direct area around the layer house

Line 306-307: Figure 5A should be Figure 5B.

Figure 5B: why is there no observation on day 25 for the layer house?

3.7. Diagnostic testing results

Line 314-324: out of curiosity, did you try to grow bacterial cultures on positive samples for Campylobacter DNA?

DISCUSSION

Line 327: “in de environment” should be “in the environment”.

Lines 349-353: it was not possible to test the chickens for the presence of Campylobacter? Do you think it would have been possible to use closed bags instead of mosquito nets, to avoid any contamination of the samples from the environment within the poultry house? This would have allowed to exclude contamination from the sentinel chickens, but I am not sure whether the sampling of outside insects and PM would have been as efficient.

In a related comment, do you think you could have missed small, potentially infected PMs (i.e. smaller than the 1.4 x 1.6 mm opening of the mosquito nets)?

Line 369: is up to a dozen of visits per day really a considerable number?

Line 373: “strictly” instead of “strict”

Line 394: remove “after” between “by” and “ingestion”

Line 394-398: what about dust and feather particles? Could this be a potential source of introduction, possibly less visible than whole feathers and more difficult to detect?

Maybe it would be interesting to discuss that air-borne transmission might be more frequent from neighbouring infected farms than from wild birds in other contexts than the Netherlands. This would emphasize your recommendations to prevent the entry of PM and insects, as it could not only prevent contamination from the surrounding environment and wild birds, but also from neighbouring infected farms, especially in densely populated poultry areas.

You may want to consider discussing the potential impact of your results on the risk of Campylobacter introduction, besides being a proof of concept of potential pathogen introduction via PM in air-inlets. Has introduction of Campylobacter in poultry farms via this transmission route been described before? Do you think it is a major transmission route of this pathogen, and what could be the implications for animal and public health?

Apart from saying that the surroundings of the farms should be considered as contaminated because of the presence of Anseriformes and Charadriiformes, you don’t really discuss the results of the camera observations, especially in relation to the results on PM introduction. Maybe one interesting aspect of your results is that, despite the substantial number of wild bird observations in the immediate surroundings of the farms, you observed very few faecal matter and no feather in the mosquito nets. You may want to discuss this aspect. Otherwise, I feel like the video-camera monitoring stands out, almost as a separate study, and that you don’t really use the results.

Reviewer 2 Report

The study is well-designed and the subject relevant. Strict and permanent biosecurity is needed to prevent the introduction of infectious agents into a flock and to prevent inter-farm transmission. The air-borne particle matter and insects may potentially transfer HPAIv or other pathogens through air-inlets into poultry houses. A wind-break mesh netting of 1.4 x 1.6 mm mesh size was used on the air-inlet outlet. 

However, minor questions have arisen due to misinterpretations and require additional edition, for an improved presentation.

Page 4 line 154 onwards 

Suggestions for improvement

... type, as artificial material (litter, such as  fabric fiber, glass, metal, plastic, ....) or natural inanimate (cobweb, feather, wood, seed, leaf, polen) or animate material such as animal (arthropod - arachnid, insect; mollusk, etc.) ...  

Are domestic flies (Musca domestica) a problem in the region? Any found ? If not found, considerations could be useful and made regarding to such negative results ?

Chewing-lice "Mallophaga" (Insecta: Phthiraptera) or mites (acarid)  (Arachnida: Acarina) species might be sub-microscopic or microscopic and filter through the mesh. Any retained will  require a degree of magnification for performing the evaluation.

Although the mesh size would enable the complete passage of lice and mites, the potential presence of these attached to larger retained particles, such as feathers, leaves and on the body of flying insects (by phoresis).

Please refer to the attached manuscript copy for details regarding suggestions to the text.
